# A Deep Learning-Aided Automated Method for Calculating Metabolic Tumor Volume in Diffuse Large B-Cell Lymphoma

**DOI:** 10.3390/cancers14215221

**Published:** 2022-10-25

**Authors:** Russ A. Kuker, David Lehmkuhl, Deukwoo Kwon, Weizhao Zhao, Izidore S. Lossos, Craig H. Moskowitz, Juan Pablo Alderuccio, Fei Yang

**Affiliations:** 1Department of Radiology, Division of Nuclear Medicine, University of Miami Miller School of Medicine, Miami, FL 33136, USA; 2Department of Public Health Sciences, University of Miami Miller School of Medicine, Miami, FL 33136, USA; 3Department of Biomedical Engineering, University of Miami, Coral Gables, FL 33146, USA; 4Sylvester Comprehensive Cancer Center, Department of Medicine, Division of Hematology, University of Miami Miller School of Medicine, Miami, FL 33136, USA; 5Sylvester Comprehensive Cancer Center, Department of Radiation Oncology, University of Miami Miller School of Medicine, Miami, FL 33136, USA

**Keywords:** artificial intelligence, deep learning, U-Net, PET/CT, diffuse large B-cell lymphoma, metabolic tumor volume

## Abstract

**Simple Summary:**

In recent years metabolic tumor volume (MTV) has been shown to predict outcomes in lymphoma. However, the current methods used to measure MTV are time-consuming and require manual input from the nuclear medicine reader. Therefore, we aimed to develop a deep-learning-aided automated method to calculate MTV. We tested this approach in 100 patients with diffuse large B-cell lymphoma enrolled in a clinical trial cohort. We observed a high correlation between nuclear medicine readers and the automated method, underscoring the potential of this approach to integrate PET-based biomarkers in clinical research.

**Abstract:**

Metabolic tumor volume (MTV) is a robust prognostic biomarker in diffuse large B-cell lymphoma (DLBCL). The available semiautomatic software for calculating MTV requires manual input limiting its routine application in clinical research. Our objective was to develop a fully automated method (AM) for calculating MTV and to validate the method by comparing its results with those from two nuclear medicine (NM) readers. The automated method designed for this study employed a deep convolutional neural network to segment normal physiologic structures from the computed tomography (CT) scans that demonstrate intense avidity on positron emission tomography (PET) scans. The study cohort consisted of 100 patients with newly diagnosed DLBCL who were randomly selected from the Alliance/CALGB 50,303 (NCT00118209) trial. We observed high concordance in MTV calculations between the AM and readers with Pearson’s correlation coefficients and interclass correlations comparing reader 1 to AM of 0.9814 (*p* < 0.0001) and 0.98 (*p* < 0.001; 95%CI = 0.96 to 0.99), respectively; and comparing reader 2 to AM of 0.9818 (*p* < 0.0001) and 0.98 (*p* < 0.0001; 95%CI = 0.96 to 0.99), respectively. The Bland–Altman plots showed only relatively small systematic errors between the proposed method and readers for both MTV and maximum standardized uptake value (SUVmax). This approach may possess the potential to integrate PET-based biomarkers in clinical trials.

## 1. Introduction

Diffuse large B-cell lymphoma (DLBCL) is the most common histologic subtype of non-Hodgkin lymphomas, with an estimated incidence of 150,000 new cases annually worldwide [1,2,3]. DLBCL is a curable disease in nearly 60% of patients treated with anthracycline-containing immunochemotherapy such as rituximab, cyclophosphamide, doxorubicin, vincristine, and prednisone (R-CHOP) and dose-adjusted etoposide, prednisone, vincristine, cyclophosphamide, doxorubicin, and rituximab (EPOCH-R) [4,5]. Patients with refractory DLBCL, however, demonstrate poor outcomes, with a median overall survival of only 6.3 months [6]. Therefore, the early identification of patients at risk for treatment failure remains a critical need in an effort to consider alternative treatment strategies in this population. 

Prognosis in patients with DLBCL is commonly determined by the International Prognosis Index (IPI) score comprised of clinical and laboratory variables [7]. The IPI score was developed in the early 1990s, undergoing subsequent validations and revisions associated with better risk assessment [8,9]. However, significant advances in the understanding of disease biology that occurred over the last two decades uncovered substantial molecular heterogeneity and associated divergent survival, which was not fully captured in the IPI score [10,11]. Furthermore, this index is not included in the treatment selection of frontline or subsequent lines of therapy, underscoring the need to develop biomarker-driven therapies in patients with DLBCL.

^18^F-fluorodeoxyglucose (FDG) positron-emission tomography with computed tomography (PET/CT) is routinely incorporated in clinical practice for the staging and assessment of treatment response in DLBCL [1,12,13,14]. The Lugano classification criteria is the most commonly used staging system for the evaluation of treatment efficacy for established and experimental therapies [15]. Metabolic tumor volume (MTV) calculated from FDG-PET/CT has been shown to be a robust prognostic biomarker across different lymphomas [16,17,18]. In patients with DLBCL, MTV demonstrated prognostication in the frontline and relapsed settings [19,20,21]. Investigators from the SAKK38/07 trial developed a prognostic model, including mutation profiling and baseline FDG-PET/CT metrics, in patients enrolled in the study. Patients with high MTV and metabolic heterogeneity demonstrated the highest risk of relapse [22]. Furthermore, Mikhaeel et al. recently developed the International Metabolic Prognostic index integrating MTV with individual components of the IPI score, such as age and stage, enabling individualized estimates of patient outcome [23]. Therefore, the implementation of MTV in clinical practice is expected to be imminent. 

Despite encouraging prognostication defined by MTV, several challenges remain for its broad implementation. Calculating MTV can be tedious and time-consuming when using currently available semiautomatic software [24]. There can also be inherent variability in calculating MTV that requires manual input from the readers [25,26,27,28,29]. The goal of the present study was to develop a fully automated method for calculating MTV. We first explored the feasibility of a fully automated method (AM) to calculate MTV in a clinical trial dataset and, subsequently, we compared the results obtained by the AM with the results obtained by two blinded readers. The contributions of our study include:Developing a novel fully automated machine learning approach for MTV calculation in DLBCL.Validating the developed approach against experienced nuclear medicine readers in determining MTV and maximum standardized uptake value (SUVmax).Enabling the integration of a machine learning approach in DLBCL clinical research.

## 2. Materials and Methods

### 2.1. Study Cohort

The clinical trial cohort consisted of 491 eligible patients with newly diagnosed DLBCL who were enrolled in the Alliance/CALGB 50,303 (NCT00118209) trial, an intergroup, randomized phase III study aimed to compare six cycles of dose-adjusted EPOCH-R with standard R-CHOP as a frontline therapy for DLBCL [30]. Eligible patients included untreated DLBCL confirmed by central pathology review. Before enrollment, limited field radiation or fewer than 10 days of glucocorticoid treatment for urgent disease complications were allowed. Additional eligibility included age ≥ 18 years, stage II to IV DLBCL (stage I primary mediastinal B-cell lymphoma was allowed), Eastern Cooperative Oncology Group performance status 0 to 2, and acceptable cardiac, renal, hematological, and liver function. The presence of central nervous system involvement and human immunodeficiency virus infection represented exclusion criteria. In the Alliance/CALGB 50,303 study dose-adjusted EPOCH-R was more toxic and did not improve progression-free survival or overall survival compared with standard R-CHOP [30,31]. Among those 491 patients, 155 whole-body FDG-PET/CT scans at study enrollment were publicly available at The Cancer Imaging Archive (TCIA) [32]. We randomly selected 100 patients to analyze for the present study.

### 2.2. Imaging Data

Imaging examinations of the selected patients were acquired from three different types of PET/CT scanners including Siemens Biograph (Siemens Medical System, Erlangen, Germany), Philips GEMINI (Philips Healthcare, Best, The Netherlands), and GE Discovery (General Electric Co., Milwaukee, WI, USA). As per the trial protocol, after confirming plasma glucose level <200 mg/dL and at least a 4-h fasting period, patients were intravenously injected with 8–20 mCi of FDG and PET/CT scans were obtained approximately 60 to 80 min afterward. Concomitant low-dose CTs, extending mainly from the skull base to thighs for anatomic localization and attenuation correction, were performed at 110–140 kVp with a reference dose of 200 mAs and iteratively reconstructed with a slice thickness ranging from 2 mm to 4 mm. PET scans were reconstructed using algorithms ranging from ordered-subset expectation maximization (OSEM) to blob-based iterative time-of-flight (BLOB-OS-TF) to point spread function (PSF) modeling with and without time-of-flight (PSF-TF). PET scan slice thickness ranged from 2 mm to 4.25 mm, with the most typical being 3.25 mm or 4.25 mm (83%). In addition, 50 whole-body CT scans from the TCIA collection of the whole-body FDG-PET/CT dataset [33] were used to fine-tune the employed deep-learning-based segmentation model. Imaging parameters of these CT scans were as follows: tube voltage of 120 kV, reference dose of 200 mAs, and slice thickness of 2–3 mm. Contours of the brain, heart, kidneys, and bladder were provided by a consensus exercise of two expert radiologists. The local institutional review board (IRB) waived the study from review as only publicly available aggregated patient datasets were utilized.

### 2.3. Segmentation of Anatomic Structures with Physiologic FDG Avidity

Anatomic structures with avid physiologic FDG uptake, such as the brain, heart, kidneys, and bladder, complicate the interpretation of PET imaging data for MTV determination. To alleviate this, a deep convolutional neural network model was deployed to segment these structures on the CTs. The segmentation model was built off the pre-trained 2D dilated residual U-net architecture by Manteia Medical Technologies (Milwaukee, WI, USA) [34]. Residual U-net was adopted due to its ability to alleviate the vanishing gradient problem as the depth of the network increases. Figure 1 illustrates the network architecture of the deployed model. Both the encoder and decoder were composed of five cascades of residual blocks. In addition, a shortcut connection was implemented between the corresponding feature maps between the encoder and decoder. Each residual block was composed of two convolution layers, and the size of the convolution kernel was 3 × 3. Each residual block was cascaded with the down-sampling layer or the upper-sampling layer. The down-sampling method used was maximum pooling and the upper-sampling method was the bilinear interpolation. Furthermore, batch normalization was also applied to reduce the internal covariate shift [35].

To fine-tune the pre-trained model towards the purpose of this work, the weights of the final output layer of the original model were reset to random values, resulting in a total of 165 trainable parameters. The dataset used for fine-tuning the pretrained model comprised the aforementioned 50 whole-body CTs annotated for the brain, heart, kidneys, and bladder, which were divided at the ratio of 5:1:4 for training, validation, and testing sets, respectively. Data preprocessing included clipping image intensity to 1–99% of the maximum and Z-Score standardization. The modified model was trained with a maximum number of training epochs of 100. The learning rate was initialized as 3 × 10^−4^ and decreased to 3 × 10^−6^ after about 60 epochs. Regarding data augmentation for training, techniques based on affine transforms such as rotation, translation, scaling, and flipping were employed. The objective function was a combination of cross-entropy and Dice loss, and adaptive moment estimation (ADAM) was utilized to update the parameters with a weight decay of 1 × 10^−4^. Training loss went from 1.3317 to 0.0190, from 1.4233 to 0.0551, from 1.2526 to 0.0774, and from 1.6453 to 0.0576 for the brain, heart, kidneys, and bladder, respectively. Training accuracy by the Dice coefficient for the brain, heart, kidneys, and bladder were 0.9885, 0.9441, 0.9145, and 0.9045, respectively. Testing accuracy by the Dice coefficient for the four target organs was 0.9524, 0.9023, 0.9107, and 0.8809, respectively. Regarding the implementation environment for the described fine-tuning process, PyTorch (v1.10) [36] was employed.

Upon being obtained on the CTs, contours of the above-mentioned FDG avid structures were transferred to the PET scans with automatic adjustment for their respective PET presentations by the aid of an array of ad hoc image-processing algorithms including region-growing, active contours, and fast matching [37,38,39] (Figure 2).

### 2.4. Automated Determination of MTV on FDG-PET

Prior to MTV calculation, a narrow trapezoid-shaped zone was established based on PET-adapted kidney and bladder contours. The zone extended in the cranial–caudal direction from the superior poles of the kidneys to the central cross-sectional plane of the bladder, in the anterior–posterior direction from the anterior to the posterior surfaces of the kidneys on the top base while on the bottom base from the anterior to the posterior borders of the bladder, as shown in the central cross-sectional plane, and in the left-right direction between the midlines of the two kidneys on the top base while, on the bottom base, between the lateral borders of the bladder, a central cross-sectional plane is shown. The rationale for creating such a zone was to aid in the identification of focal uptake by the ureters, which, incidentally, posed a challenge to the employed deep learning-based segmentation model given both the paucity of accurate training data and the wide anatomical variation of the ureters. In addition, establishing such a zone was also of help in the detection of isolated and scattered areas of focal uptake resulting from the kidneys and bladder. 

The MTV determination was conducted within the volume defined by the PET-imaged whole-body volume excluding the aforementioned anatomical structures being transferred and adapted to PET scans, including the brain, heart, kidneys, and bladder. This volume was determined by a threshold with respect to 41% of the SUVmax, [40] followed by clustering of the contiguous supra-threshold voxels into isolated regions under an additional constraint of retaining only the ones with size greater than 1 cm^3^. This resulted in the formation of a set of candidate lesion regions of interest (ROI), which was then further screened for exclusion of the ones with size less than 2 cm^3^ as well as those falling in the defined trapezoid-shaped zone. In scenarios where the candidate lesion ROI with the SUVmax was screened out, its volume was removed from the defined MTV analyzing space, and the process was repeated, until all the criteria laid out above were met. Of note, the whole described process was automatic, without requiring any manual intervention.

### 2.5. Semiautomatic Method for MTV Measurement

All FDG-PET/CT images were independently reviewed using the Hermes Affinity Viewer by two experienced nuclear medicine readers. ROIs selected by the software were manually adjusted in three planes to exclude adjacent physiologic FDG avid structures. SUVmax was defined as the maximum voxel intensity within the volumetric region of interest. Bone marrow involvement was only included in volume measurement if there was focal uptake. The spleen was considered as involved if there was focal uptake or diffuse uptake higher than 150% of the liver background. MTV was obtained by summing the metabolic volumes of all individual lesions using the previously reported 41% of SUVmax threshold and volume ≥1 cm^3^. Nuclear medicine readers were blinded for the automated results and vice versa. 

### 2.6. Statistical Analysis

MTV and SUVmax were compared to the fully automated results from the developed algorithm. To examine agreement, we estimated Pearson’s correlation coefficients and inter-class correlation coefficients (ICCs), along with corresponding 95% confidence intervals and p-values. For visualization, we displayed scatter plots along with regression lines and Bland–Altman plots between readers and the automated method. All tests were two-sided and statistical significance was considered when *p* < 0.05. Statistical software R was used for all statistical analyses.

## 3. Results

We sought to investigate the performance of a three-dimensional deep learning-aided AM for MTV calculation in 100 patients with DLBCL enrolled in the Alliance/CALGB 50,303 clinical trial. There were 17 centers participating in this trial and the PET/CT systems employed included: Siemens (*n* = 53), GE (*n* = 30), and Philips (*n* = 17). Among the randomly selected patients, the mean MTV calculated by reader 1 was 226.470 mL (standard deviation (SD) 260.066 and coefficient of variation (CV) 114.834), for reader 2 was 226.799 mL (SD 261.965 and CV 115.505) and for AM was 205.704 mL (SD 245.825 and CV 119.504). 

Comparing reader 1 to reader 2, the Pearson’s correlation coefficients and ICCs were 0.9997, *p* < 0.0001 and 1, *p* < 0.0001 (95%CI = 1 to 1) for MTV and 1, *p* < 0.0001 and 1, *p* < 0.0001 (95%CI = 1 to 1) for SUVmax, respectively (Figure 3A,B). Comparing reader 1 to AM, the Pearson’s correlation coefficients and ICCs were 0.9814, *p* < 0.0001 and 0.98, *p* < 0.0001 (95%CI = 0.96 to 0.99) for MTV and 0.9868, *p* < 0.0001 and 1, *p* < 0.0001 (95%CI = 0.99 to 1) for SUVmax, respectively (Figure 3C,D). Comparing reader 2 to AM, the Pearson’s correlation coefficients and ICCs were 0.9818, *p* < 0.0001 and 0.98, *p* < 0.0001 (95%CI = 0.96 to 0.99) for MTV and 0.9868, *p* < 0.0001 and 1, *p* < 0.0001 (95%CI = 0.99 to 1) for SUVmax, respectively (Figure 3E,F).

When we assessed the data sorted by the type of PET/CT system, we observed small differences in SUVmax between the readers and AM only on images obtained by Philips scanners (readers and AM: ICC 0.81, *p* < 0.0001 (95%CI = 0.57 to 0.93)) (Appendix A). We did not observe differences by the type of scanner in MTV volumes. (Appendix A). 

The Bland–Altman plots showed only relatively small systematic errors between the proposed method and the manual readings across the entire data range being examined for both MTV (Figure 4) and SUVmax (Figure 5). 

Subsequently, we calculated the Root-Mean-Squared Error (RMSE) between readers (average) and the proposed AM as a measure of accuracy and positive difference and negative difference between the two measurements as a bias. For MTV calculations, the RMSE was 54.7, with a positive bias of 28.4 and a negative bias of 0.27 (Appendix A). The mean difference between readers was 20.92 (95% limits of agreement of −49.77 and 91.63). AM demonstrated smaller MTV values compared to those of the nuclear medicine readers. For SUVmax calculations, we found an RMSE of 1.93 with a positive bias of 15.4 and a negative bias of 1.26 (Appendix A). The mean difference between readers was −0.03 (95% limits of agreement of −3.34 and 3.26). Again, AM demonstrated smaller values of SUVmax compared to the nuclear medicine readers.

## 4. Discussion

In this study, we showed that a deep-learning-aided method can accurately segment lymphoma lesions, allowing for a fully automated assessment of MTV in a homogeneously treated patient population. SUVmax and tumor volumes measured by our proposed method were highly correlated with those determined by independent readers using a semiautomatic software, validating these results. No subjects were excluded due to failure of the automated method. Furthermore, the algorithm was highly accurate in classifying FDG-avidity in patients from a multicenter clinical trial involving 17 centers that obtained images on different scanner models with variable reconstruction settings. 

Deep learning is a subtype of representation learning aimed to describe complex data representations using simpler hierarchized structures defined from a set of specific features [41]. Convolutional neural networks represent the core of deep learning methods for imaging and are multilayered artificial neural networks with weighted connections between neurons that are iteratively adjusted through repeated exposure to training data. These networks may be used for the automation of various time-consuming tasks including image detection, segmentation, and classification [42]. This method possesses the potential to decrease reading time and increase the reproducibility of measurements and has been associated with similar accuracy to semiautomatic methods that require reader input [43,44,45]. 

The availability of predictive factors of response to standard and experimental regimens remains an unmet need in DLBCL. More recently, several automated segmentation methods have been proposed in DLBCL [45,46,47,48,49]. Capobianco et al. examined a machine learning approach to generate MTV in DLBCL [47]. The authors tested an investigational software prototype (PET-Assisted Reporting System (PARS); Siemens Medical Solutions USA, Inc., Malvern, PA, USA) to estimate MTV in 301 patients enrolled in the REMARC clinical trial [47,50]. The automated whole-body high-uptake segmentation algorithm identified all three-dimensional regions of interest with increased tracer uptake. The resulting ROIs were processed using a convolutional neural network trained on an independent cohort. They observed a similar correlation between PARS-based MTV with reference MTV calculated by two experienced readers (ρ = 0.76; *p* < 0.001). Subsequently, Jiang et al. trained a 3-D U-Net architecture on patches randomly sampled within PET images in 414 patients with DLBCL [48]. Authors found a strong positive correlation (linear regression analysis; R^2^ linear = 0.882, *p* < 0.001) between ground-truth MTV and predictive MTV in training and validation (R^2^ linear = 0.939, *p* < 0.001) cohorts. Most recently, Revailler et al. completed a training dataset of 407 patients in 93 h underscoring the speed of current deep-learning models to compute MTV [45]. 

The automated method proposed here brings a new solution to the problem of MTV calculation in DLBCL and has several advantages compared to the previous methods. First, when compared to the previous methods, which are more or less “black box” models that are difficult to interpret and often provide little insight into how decisions are made, the proposed method is more explicit and more direct in emulating how nuclear medicine physicians reason through DLBCL PET/CT imaging data. Moreover, the inherent human bias induced by inter- and intra-observer perception errors in reading PET/CT scans for MTV calculation is eliminated by the proposed method since it does not need the massive quantities of annotated training data on which others rely. In addition, the proposed method with the use of segmentation of physiologic FDG avid structures on CTs may be advantageous for patient cases featuring a low tumor burden, for which the previous methods are particularly problematic.

Limitations of the present study include the applicability of our results to other lymphoma subtypes and cancer groups and the need to further validate and refine our automated method. Although our sample size is relatively small, patients were randomly selected from a homogeneous dataset, and we observed similar results across our cohort. Furthermore, the presented performance of the developed method should be interpreted with caution, given that the method was validated against readings collected from only one, although generally accepted and widely used, dedicated semiautomatic MTV calculation software. In addition, the manual readings for this study were performed by readers from the same institution, which may lend itself to potential reader bias. We did not seek to develop a predictive or prognostic model due to the incomplete availability of PET/CT scans from TCIA. Our goal was limited to validating our automated method approach. Finally, the performance of the proposed automated MTV calculation method may deteriorate in some rare but complicated clinical scenarios, such as tumor activity being located in close proximity to normal physiologic structures such as the bladder or kidneys, or when normal anatomy is distorted either due to the disease process or image artifacts, including misregistration or patient motion amongst others. 

Nonetheless, the proposed automated method is strengthened by its ability to calculate MTV with a high correlation to analysis by expert readers in the company of automation and high throughput (median process time: 5 min for the proposed method vs. 20 min for expert analysis). Developing a fully automated method, such as ours, for calculating MTV that is accurate and reproducible may facilitate the application of MTV in clinical research, providing real-time risk stratification. Future studies should prospectively explore treatment decisions based on MTV data. 

## 5. Conclusions

We demonstrated that a deep-learning-aided, fully automated method is capable of calculating MTV in patients with DLBCL. The resulting MTV values were highly concordant with the results obtained by two blinded nuclear medicine readers. Employing deep learning for the calculation of MTV offers many advantages over semiautomated methods, including time efficiency and the reproducibility of results across different PET/CT systems. The proposed automated method is unique in that it emulates how nuclear medicine readers analyze PET/CT images and does not require massive quantities of annotated training data. We believe that an accurate and highly reproducible automated method for calculating MTV has great potential for incorporation into clinical research.

## Figures and Tables

**Figure 1 cancers-14-05221-f001:**
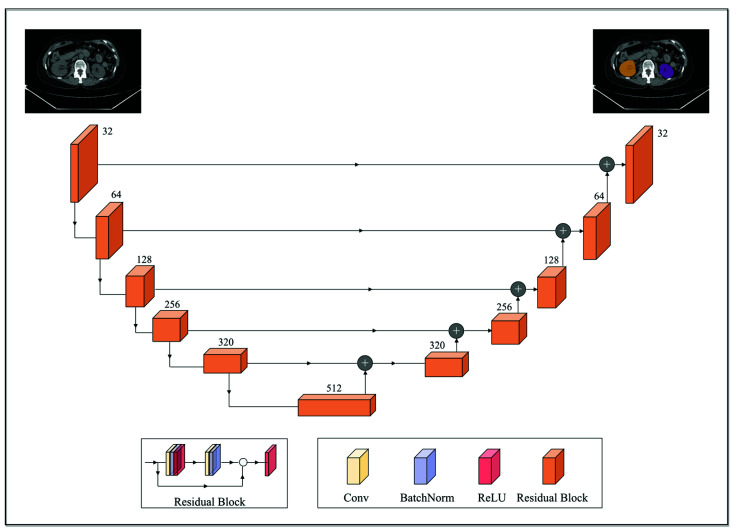
A schematic overview of the employed 2D dilated residual U-net-based segmentation model. The encoder and decoder were composed of 5 cascades of residual blocks. Each residual block was composed of two convolution layers and was cascaded with the downsampling layer (maximum pooling; down arrow) or the upper sampling layer (bilinear interpolation; upper arrow). A shortcut connection (horizontal arrow) was implemented between the corresponding feature maps between the encoder and decoder.

**Figure 2 cancers-14-05221-f002:**
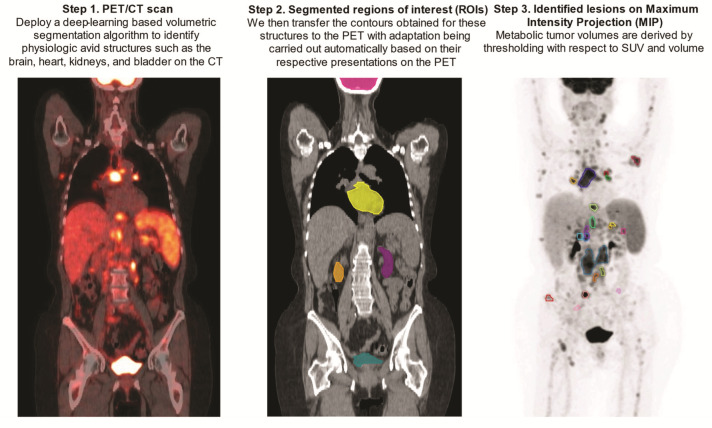
Step-by-step demonstration of deep-learning-aided metabolic tumor volume calculations.

**Figure 3 cancers-14-05221-f003:**
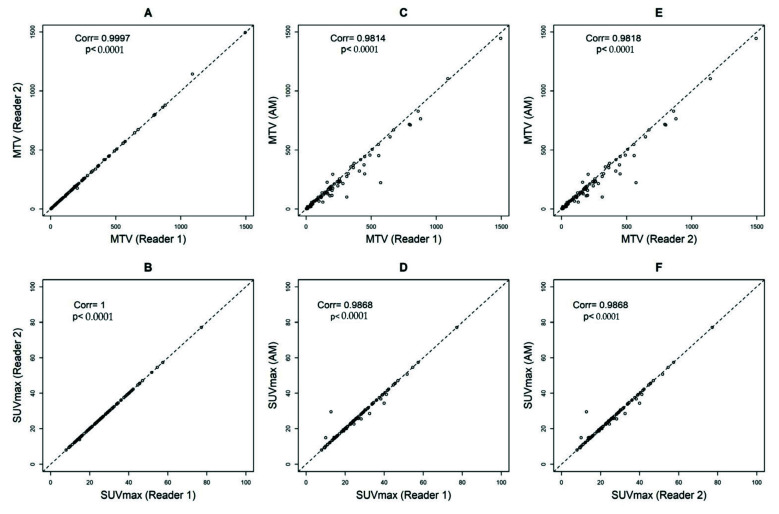
Pearson’s correlation coefficients calculating metabolic tumor volumes (MTV) with a threshold of 41% and SUVmax between Reader 1 and Reader 2 (**A**,**B**), Automated Method (AM) approach and Reader 1 (**C**,**D**), and AM and Reader 2 (**E**,**F**).

**Figure 4 cancers-14-05221-f004:**
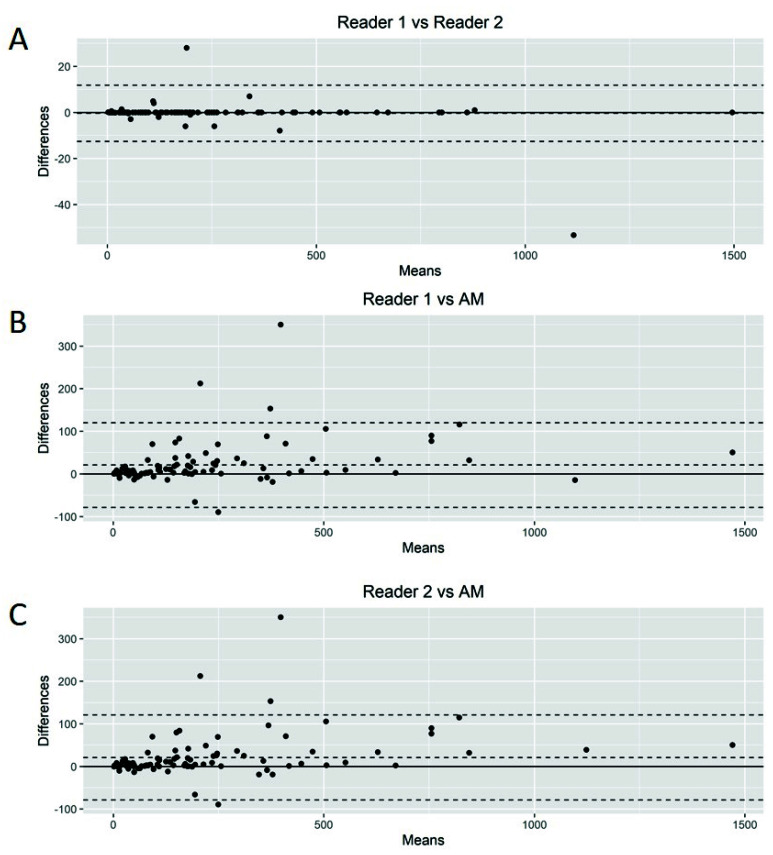
Bland–Altman plot. Graphical display for bias between two readers and automated method (AM) in metabolic tumor volume calculation (**A**–**C**).

**Figure 5 cancers-14-05221-f005:**
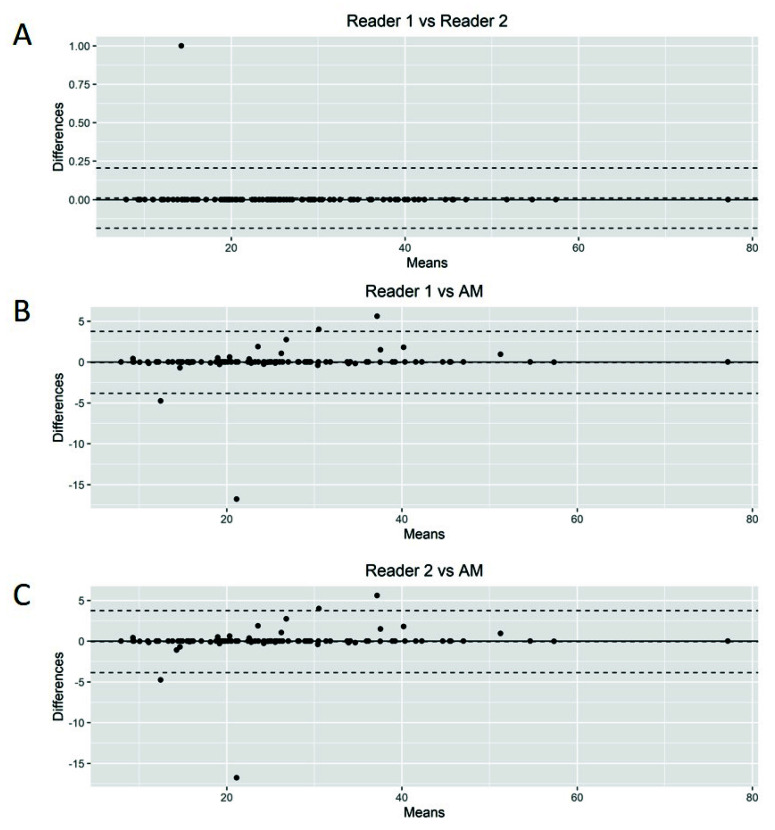
Bland–Altman plot. Graphical display for bias between two readers and automated method (AM) in SUVmax calculation (**A**–**C**).

## Data Availability

The Cancer Imaging Archive is a service which de-identifies and hosts a large archive of medical images of cancer accessible for public download. https://www.cancerimagingarchive.net/ (accessed on 1 October 2021).

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
