# Peer review of "A Deep Learning-Aided Automated Method for Calculating Metabolic Tumor Volume in Diffuse Large B-Cell Lymphoma"

_cancers, 2022, doi:10.3390/cancers14215221_

Round 1

Reviewer 1 Report

The overall impression of the technical contribution of the current study is Promising. However, the Authors may consider doing necessary amendments to the manuscript for better comprehensibility of the study.

Manuscript Organization and Formatting:

1.     Few of the abbreviations are not clearly presented. Please expand the abbreviations only on their first occurrence. For example: SUV is not abbreviated and MTV is expanded twice.

Similarly “DLBCL” need not be expanded as it's already expanded in the abstract.

2.     Keywords do not seem timely, kindly update them for a better searching experience of the manuscript.

3.     The contribution of the current study must be briefly discussed as bullet points in the introduction. And motivation must also be discussed in the manuscript.

4.     The overall organization of the manuscript is not discussed anywhere in the manuscript. Please add the same in the introduction section of the manuscript.  

5.     Please make the study evident with an adequate number of references, there are few statements observed without references to endorse the claim.

For example: The statement “Concomitant low-dose CTs extending mainly from the skull…”

6.     In the architecture presented in figure 1, authors may consider presenting the size of the input CT/PET scan images like 64 x 64/128 x 128.

7.     Please discuss more on the implementation platform and the dataset details as two sub-sections in the manuscript.

8.     By considering the current form of the conclusion section, it is hard to understand by Journal readers. It should be extended with new sentences about the necessity and contributions of the study by considering the author's opinions about the experimental results derived from some other well-known objective evaluation values if it is possible.

9.      Authors should use more alternative models as the benchmarking models, authors should also conduct some statistical tests to ensure the superiority of the proposed approach, i.e., how could authors ensure that their results are superior to others? Meanwhile, the authors also have to provide some insightful discussion of the results.

10.  English proofreading is strongly recommended for a better understanding of the study.

Technical Aspects

1.     The work is interesting but not obviously novel, authors must justify how the proposed model is better than the existing one.

2.     What is the learning rate of the proposed model. (both initial and learning rate at saturation)

3.     Where are the hyperparameters discussed in the study. Training accuracy, testing accuracy, training loss, and testing loss.

4.     Where is the pooling layer discussed in the current architecture. More adequate discussion must be presented in the manuscript.

5.     Does the authors perform any batch normalization.

6.     From the statement, “statistical significance was considered when p<.05.”, why the value 0.5 is considered.?

7.     What is R2 measure from line 242 of manuscript.

8.     From the statement, “compared to a semiautomatic approach”, what are those semiautomatic approaches considered in the current study for analysis.

9.     What is the systematic error observed in the current study.

10.   Any overall accuracy of the model from the current study?

Author Response

Reviewer 1 Comments: 

Manuscript Organization and Formatting:

1) Few of the abbreviations are not clearly presented. Please expand the abbreviations only on their first occurrence. For example: SUV is not abbreviated, and MTV is expanded twice. Similarly, “DLBCL” need not be expanded as it's already expanded in the abstract.

Response: We thank the reviewer for noticing these errors. However, we believe the reviewer is pointing out the expansion of MTV abbreviation located in Simple Summary and the Abstract. As these are two independent sections, we believe remains appropriate to expand MTV twice. Nevertheless, we thoughtfully reviewed the manuscript and all abbreviations have been expanded only at first occurrence.

2) Keywords do not seem timely, kindly update them for a better searching experience of the manuscript.

Response: Keywords were updated as suggested. Please see page 1 of the edited manuscript.

3) The contribution of the current study must be briefly discussed as bullet points in the introduction. And motivation must also be discussed in the manuscript.

Response: In the edited manuscript we now include the contribution of this manuscript. Please see Page 2 and 3 of edited manuscript.

4) The overall organization of the manuscript is not discussed anywhere in the manuscript. Please add the same in the introduction section of the manuscript.

Response: Overall organization is now included in the introduction of edited manuscript. Please see Page 2 of edited manuscript.

5) Please make the study evident with an adequate number of references, there are few statements observed without references to endorse the claim. For example: The statement “Concomitant low-dose CTs extending mainly from the skull…”

Response: New references have been added as suggested by the reviewer.

6) In the architecture presented in figure 1, authors may consider presenting the size of the input CT/PET scan images like 64 x 64/128 x 128.

Response: We thank the reviewer for this suggestion. The input images and output segmentation map are now added to edited Figure 1.

7) Please discuss more on the implementation platform and the dataset details as two sub-sections in the manuscript.

Response: Implementation platform and dataset details sections have been expanded.  On edited manuscript these data appear as two sub-sections in edited page 3.  

8) By considering the current form of the conclusion section, it is hard to understand for Journal readers. It should be extended with new sentences about the necessity and contributions of the study by considering the author's opinions about the experimental results derived from some other well-known objective evaluation values if it is possible.

Response: We expanded conclusion section based on reviewer’s comment. Please see edited page 11.

9) Authors should use more alternative models as the benchmarking models, authors should also conduct some statistical tests to ensure the superiority of the proposed approach, i.e., how could authors ensure that their results are superior to others? Meanwhile, the authors also have to provide some insightful discussion of the results.

Response: We thank the reviewer for this comment. Our analysis focuses on assessing the concordance between three readers (2 nuclear medicine physicians and a machine learning approach) and not in the development of a model. All three readers are not significantly different from each other since all confidence intervals are overlapped. Therefore, we cannot conduct a statistical analysis comparing with alternative models.

10) English proofreading is strongly recommended for a better understanding of the study.

Response: English proofreading was performed across the manuscript.

Technical Aspects:

1) The work is interesting but not obviously novel, authors must justify how the proposed model is better than the existing one.

Response: We thank the reviewer for the positive comments regarding this work of ours. As to the advantages of the proposed method over the existing ones, we have highlighted the following in the revised manuscript: (i) when compared to the existing methods, which are more or less "black box" models that are difficult to interpret and oftentimes provide little insight into how decisions are made, the proposed method is more explicit and more direct in emulating how nuclear medicine physicians reason through DLBCL PET/CT imaging data; (ii) the inherent human bias induced by inter- and intra-observer perception errors in MTV calculation is eliminated by the proposed method as it does not need the massive quantities of annotated training data that the existing ones heavily rely on; (iii) the proposed method with the use of segmentation of physiologic FDG avid structures on CTs may be beneficial in MTV calculation for patients with low tumor burden, for which the existing methods are reportedly especially problematic. A discussion with regard to this matter has been added in the Discussion section of the revision. Please see Line 313-323 on page 10.

2) What is the learning rate of the proposed model. (both initial and learning rate at saturation)

Response: We thank the reviewer for pointing out this important omission. The learning rate was initialized as 3×10-6 and grew linearly to 3×10-4. A comment on this is added for clarification. Please see Line 154-155 on page 4.

3) Where are the hyperparameters discussed in the study. Training accuracy, testing accuracy, training loss, and testing loss.

Response: We apologize for the oversight and have now provided the training accuracy, testing accuracy, training loss, and testing loss. Please see Line 157-162 on page 4.

4) Where is the pooling layer discussed in the current architecture. More adequate discussion must be presented in the manuscript.

Response: Each residual block was cascaded with the down sampling layer (maximum pooling; down arrow in Figure 1) or the upper sampling layer (bilinear interpolation; upper arrow in Figure 2). We apologize for not being clear with respect to this matter in the original submission. A comment in relation to this is added to Figure 1 caption for clarification.

5) Does the authors perform any batch normalization.

Response: Of course. We thank the reviewer for having spotted this omission; apologies about this. A comment is added for clarification, please see Line 146-147 on page 4.

6) From the statement, “statistical significance was considered when p<.05.”, why the value 0.05 is considered?

Response: We thank the reviewer for highlighting this. A p-value of 0.05 or less is an arbitrary but commonly used criterion for determining whether or not an observed difference is “statistically significant”. It implies a 5% or lower likelihood that the observed differences were the result of chance, which calls for rejecting the null hypothesis and accepting the alternative hypothesis. In addition, the analyses being conducted in the present study did not involve multiple comparisons and multiple testing and thus no adjustment to the p-value threshold was either needed. We hope these explanations clarify the reviewer’s concern.

7) What is R2 measure from line 242 of manuscript.

Response: We apologize for this omission. The name of the analysis method has been added for clarification. Please see Line 307 page 10.

8) From the statement, “compared to a semiautomatic approach”, what are those semiautomatic approaches considered in the current study for analysis.

Response: The semiautomatic approach was meant to refer to the semiautomatic method being employed in the current study, in which MTV was calculated by expert nuclear medicine readers using the Hermes Affinity Viewer to analyze FDG-PET/CT images. We apologize for not being clear with respect to this matter in the original submission, and text has been rephrased to avoid confusion. Please see Line 334-336 on page 11.

9) What is the systematic error observed in the current study.

Response: please see below response addressing comments #9 and #10. 

10) Any overall accuracy of the model from the current study?

Response: To address reviewer’s comment, we calculated the Root-Mean-Squared Error (RMSE) between readers (average) and proposed automated method as a measure of accuracy and positive difference and negative difference between two measurements as a bias (shown in Bland-Altman plots in Supplemental Figures 1A and B). For SUVmax calculations we found an RMSE of 1.93 with positive bias of 15.4 and negative bias of 1.26. The automated method demonstrated to have smaller values compared to those of a nuclear medicine reader. 

For MTV calculations, the RMSE was 54.7 with a positive bias of 28.4 and negative bias of 0.27. Again, the automated method demonstrated smaller values compared to nuclear medicine reader.  

Reviewer 2 Report

This is the review report of the paper which is titled

 A deep learning-aided automated method for calculating metabolic tumor volume in diffuse large B-cell lymphoma “.

The paper has some issues that need to be addressed.

1-      The used dataset is very small which raises the concern of overfitting, I would suggest testing the trained model on an independent test set to prove the opposite.

2-         In terms of residual U-net, what is its novelty about it?

3-         More effort on the overall novelty of the paper. It is straight forward implementation.

4- Clear sentences about the research problem, in the other words, what is the problem of previous methods that this paper solved?

5-         More details about the deep learning part, training parameters, and feature visualization.

6-         Details about the used dataset with how it was divided for training, validation, testing, with ratio.

7- comparison with previous methods is necessary to add to the recent dataset.

8- More recent papers should be added.

9- More effort is required to present the manuscript in a good way.

Author Response

Reviewer 2 Comments:

The paper has some issues that need to be addressed.

1) The used dataset is very small which raises the concern of overfitting, I would suggest testing the trained model on an independent test set to prove the opposite.

Response: The approach we implemented in the current study differs to prior studies. Both nuclear medicine physicians were blinded to volumes results calculated by the other reader and our machine learning results. Similarly, the scientist who calculated tumor volumes with machine learning was blinded to readers volumes. Acknowledging the need to confirm our results in larger future studies we believe that present results are accurate. Nevertheless, this was included in the edited manuscript as a limitation of our present manuscript.

2) In terms of residual U-net, what is its novelty about it?

Response: The novelty of residual U-net in comparison with the conventional U-net lies in its ability to alleviate, if not completely overcome, the problem of vanishing gradient. It uses a skip connection in which the original input is added to the output of the convolution block and hence can be propagated as deep as possible through the network. In so doing, it allows for deeper models and helps the network learn more complex features. A comment in relation to this matter is added. Please see Line 138-139 on page 4.

3) More effort on the overall novelty of the paper. It is straight forward implementation.

Response: We thank the reviewer for pointing this to our attention. As to the novelty of the proposed method in comparison with the existing ones, we have highlighted the following in the revised manuscript: (i) when compared to the existing methods, which are more or less "black box" models that are difficult to interpret and oftentimes provide little insight into how decisions are made, the proposed method is more explicit and more direct in emulating how nuclear medicine physicians reason through DLBCL PET/CT imaging data; (ii) the inherent human bias induced by inter- and intra-observer perception errors in MTV calculation is eliminated by the proposed method as it does not need the massive quantities of annotated training data that the existing ones heavily rely on; (iii) the proposed method with the use of segmentation of physiologic FDG avid structures on CTs may be beneficial in MTV calculation for patients with low tumor burden, for which the existing methods are reportedly especially problematic. A discussion with regard to this matter has been added in the Discussion section of the revision. Please see Line 313-323 on page 10.

4) Clear sentences about the research problem, in the other words, what is the problem of previous methods that this paper solved?

Response: Please see response to Technical Aspects question #1 of first reviewer addressing this comment. 

5) More details about the deep learning part, training parameters, and feature visualization.

Response: More details about the deep learning model used for the segmentation of the brain, heart, kidneys, and bladder have now been provided. Please see Line 148-162 on page 4.

6) Details about the used dataset with how it was divided for training, validation, testing, with ratio.

Response: We appreciate the reviewer for pointing out this important omission. The dataset used to fine-tune the segmentation model were divided at the ratio of 5:1:4 for training, validation, and testing sets, respectively. A comment in relation to this matter is added. Please see Line 149-152 on page 4.

7) Comparison with previous methods is necessary to add to the recent dataset.

Response: Please see our response to your comment #3 addressing this.

8) More recent papers should be added.

Response: More recent papers have now been included in edited manuscript.

9) More effort is required to present the manuscript in a good way.

Response: We have significantly edited the manuscript based on your excellent feedback.

Reviewer 3 Report

The authors indicate that 3 different systems were used; see line 90-92 on p.2/11:

It should be mentioned how many studies are belonging to each system (e.g. in the method section), and should be further specified if a difference between the 3 different types of PET/CT scanners has been found.

No further remarks.

Author Response

Reviewer 3 Comments:

The authors indicate that 3 different systems were used; see line 90-92 on p.2/11:

It should be mentioned how many studies are belonging to each system (e.g. in the method section), and should be further specified if a difference between the 3 different types of PET/CT scanners has been found.

Response: We thank the reviewer for this important point. Three PET/CT systems were implemented in our study and distribution was as follows: Siemens (n=53), General Electrics (n=30), and Philips (n=17). Based on reviewer’s suggestion we analyzed differences in MTV and SUVmax calculations between a nuclear medicine reader and proposed machine learning approach. We did observe differences in SUVmax calculations obtained by Philips system, however, these differences did not translate in MTV calculations. The three systems performed similarly to MTV values. These data are now included in edited manuscript in page 8-9. 

Round 2

Reviewer 1 Report

The overall impression of the technical contribution of the manuscript is Promising. However, the authors need to do minor amendments to the technical aspects for better comprehensibility of the study.

1. What is the learning rate of the proposed model, the initial learning rate, and after how many epochs does the model saturated.?

2. Totally how many epochs are being employed in the training process?

3. What is/are the number of trainable parameters associated with the current model?

4. where is the dataset description added, and where is the implementation environment discussed? 

5. what are the associated hyperparameters.? training loss and accuracy, similarly the testing loss and accuracy.

Author Response

Reviewer 1

The overall impression of the technical contribution of the manuscript is Promising.

Response: We thank the reviewer for recognizing our work.

However, the authors need to do minor amendments to the technical aspects for better comprehensibility of the study.

  1. What is the learning rate of the proposed model, the initial learning rate, and after how many epochs does the model saturated?

Response: The learning rate was initialized as 3×10-4 and decreased to 3×10-6 after about 60 epochs. We apologize for this oversight and a comment on this is added for clarification. Please see Line 154-155.

  1. Totally how many epochs are being employed in the training process?

Response: The maximum number of epoch was set to 100 in the training process. A comment on this is added for clarification. Please see Line 153-154.

  1. What is/are the number of trainable parameters associated with the current model?

Response: For the modified model, there are 33 trainable parameters for each organ and 165 trainable parameters in total for the five organs of interest. We apologize for not being clear on this matter in the prior submission. A comment is added and please see Line 147-149.  

  1. Where is the dataset description added, and where is the implementation environment discussed? 

Response: We apologize for not being clear with respect to these matters. The dataset description was added in between lines 122-127 of the revised manuscript. The implementation environment was PyTorch (v1.10), and a comment in regard to this was added. Please see Line 164-165.

  1. what are the associated hyperparameters? training loss and accuracy, similarly the testing loss and accuracy.

Response: Other associated hyperparameters consisted of those used in data augmentation including rotation, translation, scaling, and flipping for training. The objective function was a combination of cross entropy and Dice loss. Training loss went from 1.3317 down to 0.0190, from 1.4233 to 0.0551, from 1.2526 to 0.0774, from 1.6453 to 0.0576 for the brain, heart, kidneys, and bladder, respectively. Training accuracy by the Dice coefficient for brain, heart, kidneys, and bladder were 0.9885, 0.9441, 0.9145, and 0.9045, respectively. Testing accuracy by the Dice coefficient for the four target organs were 0.9524, 0.9023, 0.9107, and 0.8809, respectively. These details are now included in the revised manuscript, and please see Line 157-163.

Reviewer 2 Report

The paper is not ready yet 

Lack of novelty is the main reason 

Some training set is a big issue,  suggest to think about Transfer learning  from same domain 

More effort on the presentation of the paper 

More results to show 

Author Response

Reviewer 2 

The paper is not ready yet. Lack of novelty is the main reason. Some training set is a big issue, suggest to think about Transfer learning from same domain. More effort on the presentation of the paper. More results to show. 

Response: We acknowledge the reviewer's concerns over novelty of this work of ours being developed utilizing some of the presently established deep learning techniques. However, the application of these techniques towards a fully automated approach for metabolic tumor volume calculation in patients with diffuse large B-cell lymphoma was done, to the best of our knowledge, for the first time. The differences between our approach and others on this topic, as detailed in the edited manuscript, highlight the uniqueness and accessibility of our approach. Over and above that, our method bears direct and important clinical implications as it will enable the development of biomarker-driven clinical trials and measure metabolic tumor volumes on a large scale of patients providing significant prognostic data. As to the concern in regard to the learning of the modified deep learning-based model being employed, the Dice coefficients of the organs of interest attained greater than or close to 0.90 for both the training and testing datasets, demonstrating the model architecture is capable of providing sufficient accuracy so as to be used for the segmentation task at hand. To attempt to build another deep-learning based segmentation model from scratch would, in the way we see it, be out of scope of the current work. Nevertheless, we do appreciate the innovative spirit of the reviewer. Finally, as respects the presentation of the paper, we have significantly expanded the results section, added supplemental data, and modified the method section for readability. The critical review and suggestions from the reviewer has definitely helped to improve the manuscript, to which we once more would like to extend our sincere gratitude.

Round 3

Reviewer 2 Report

Dear Authors,

Sorry to reject the paper. I believe it is a good paper but it is not finished yet. Small training data raise the concerns of overfitting. Also,  compare your results to the state of the art.